# Preclinical Therapeutic Assessment of a New Chemotherapeutics [Dichloro(4,4’-Bis(2,2,3,3-Tetrafluoropropoxy) Methyl)-2,2’-Bipryridine) Platinum] in an Orthotopic Patient-Derived Xenograft Model of Triple-Negative Breast Cancers

**DOI:** 10.3390/pharmaceutics14040839

**Published:** 2022-04-11

**Authors:** Tzu-Chun Kan, Mei-Hsiang Lin, Chun-Chia Cheng, Jeng-Wei Lu, Ming-Thau Sheu, Yuan-Soon Ho, Sri Rahayu, Jungshan Chang

**Affiliations:** 1Graduate Institute of Medical Sciences, College of Medicine, Taipei Medical University, Taipei 11031, Taiwan; d119102016@tmu.edu.tw; 2School of Pharmacy, College of Pharmacy, Taipei Medical University, Taipei 11031, Taiwan; mhl00001@tmu.edu.tw (M.-H.L.); mingsheu@tmu.edu.tw (M.-T.S.); 3Radiation Biology Research Center, Institute for Radiological Research, Chang Gung University/Chang Gung Memorial Hospital, Taoyuan 33302, Taiwan; cccheng.biocompare@gmail.com; 4Antimicrobial Resistance Interdisciplinary Research Group, Singapore-MIT-Alliance for Research and Technology, Singapore 138602, Singapore; jengweilu@gmail.com; 5College of Medical Science and Technology, Taipei Medical University, Taipei 11031, Taiwan; hoyuansn@tmu.edu.tw; 6Department of Biology, Faculty of Mathematics and Natural Science, Universitas Negeri Jakarta, Jakarta 13220, Indonesia; srirahayu@unj.ac.id; 7International Master/Ph.D. Program in Medicine, College of Medicine, Taipei Medical University, Taipei 11031, Taiwan; 8International Ph.D. Program for Cell Therapy and Regeneration Medicine, College of Medicine, Taipei Medical University, Taipei 11031, Taiwan

**Keywords:** cisplatin, cisplatin-resistant, cell line-derived xenograft model, triple-negative breast cancers, patient-derived xenograft, apoptosis, autophagy, PD-L1

## Abstract

Cisplatin is one of the most common therapeutics used in treatments of several types of cancers. To enhance cisplatin lipophilicity and reduce resistance and side effects, a polyfluorinated bipyridine-modified cisplatin analogue, dichloro[4,4’-bis(2,2,3,3-tetrafluoropropoxy)methyl)-2,2’-bipryridine] platinum (TFBPC), was synthesized and therapeutic assessments were performed. TFBPC displayed superior effects in inhibiting the proliferation of several cisplatin-resistant human cancer cell lines, including MDA-MB-231 breast cancers, COLO205 colon cancers and SK-OV-3 ovarian cancers. TFBPC bound to DNA and formed DNA crosslinks that resulted in DNA degradation, triggering the cell death program through the PARP/Bax/Bcl-2 apoptosis and LC3-related autophagy pathway. Moreover, TFBPC significantly inhibited tumor growth in both animal models which include a cell line-derived xenograft model (CDX) of cisplatin-resistant MDA-MB-231, and a patient-derived xenograft (PDX) model of triple-negative breast cancers (TNBCs). Furthermore, the biopsy specimen from TFBPC-treated xenografts revealed decreased expressions of P53, Ki-67 and PD-L1 coupled with higher expression of cleaved caspase 3, suggesting TFBPC treatment was effective and resulted in good prognostic indications. No significant pathological changes were observed in hematological and biochemistry tests in blood and histological examinations from the specimen of major organs. Therefore, TFBPC is a potential candidate for treatments of patients suffering from TNBCs as well as other cisplatin-resistant cancers.

## 1. Introduction

Cancer is one of the most intriguing malignant diseases to cure due to the numerous variants in mutations coupled with metabolic alterations, evasion of immune surveillance and the emergence of drug resistance. According to the 2021 online report of the World Health Organization (WHO) on cancer, there were nearly 10 million patients who died of cancer, making it the leading cause of death worldwide in 2020. To eradicate these cancer cells, several therapeutic regimens, including surgery, hormone therapy, radiotherapy, hyperthermia, chemotherapy, targeted therapy [1] and immunotherapy [2], have been developed and applied in treatments of cancer patients. Among them, chemotherapy is the most versatile therapeutic strategy against various forms of cancers. Not only can it serve as the main treatment, but also as part of neoadjuvant systemic therapy or adjuvant therapy according to the therapeutic plan or demand. Many kinds of chemotherapy are developed and characterized into several categories including alkylating agents, nitrosoureas, antimetabolites, antitumor antibiotics, topoisomerase inhibitors, mitotic inhibitors, corticosteroids and other chemotherapy drugs such as all-trans-retinoic acid and hydroxyurea. They are used to treat cancers either alone or in combination with other drugs or treatments. Platinum-based antineoplastic drugs, which are alkylating agents, are used to treat almost half of individuals receiving chemotherapy for cancers [3]. Currently, there are five complexes approved and available as platinum-based anticancer drugs, and they include cisplatin, carboplatin (Paraplatin), oxaliplatin (Eloxatin), nedaplatin (Aqupla) and lobaplatin [4,5].

Cisplatin (cis-diamminedichloro-platinum (II), CDDP) is the first metal-based antitumor drug approved by Food and Drug Administration in 1978. It has served as a prototype for developing numerous metal-containing antineoplastic derivatives in the pharmaceutical industry. When cisplatin has its two chlorides severed as leaving groups, the molecule preferentially binds to the N7 reactive center on purine residues of nucleic acids, leading to DNA lesion formation that interferes with normal transcription, and/or DNA replication. Subsequently, the unsolved bulky DNA lesions trigger cell cycle arrest and death program. These cytotoxic properties of cisplatin and its derivatives allowed them to emerge as the most active chemotherapeutic drugs for treatments of various forms of cancers, including ovarian, head and neck, breast, lung and bladder cancers [6,7].

Cisplatin and platinum-based antitumor drugs are commonly used in chemotherapeutic regimes for aggressive and metastatic cancers. However, two major challenges of cisplatin include the intrinsic and acquired drug resistance coupled with various side effects, which limit their effectiveness and applications in clinical settings. Several side effects, such as ototoxicity, nausea, vomiting, neurotoxicity and nephrotoxicity, have been reported in patients after the administration of cisplatin [8,9]. These undesirable side effects in patients may abruptly interrupt chemotherapeutic cycles and cripple the treatment regimen, leading to an increased risk in resistance acquisition and recurrence. Therefore, developing a new generation of platinum-based chemotherapeutics with superior therapeutic effects and milder adverse responses for cancer patients remains an urgent topic in medicine as well as the pharmaceutical industry.

2,2′-bipyridine (bpy) is a bidentate chelating ligand with many transition metals, such as gold, (III) [10,11], copper (II/I) [12], zinc (II) [13], ruthenium (Ru) and platinum (II) [14,15,16], due to its robust redox stability and ease of functionalization. Ever since the synthesis and spectroscopic studies of [platinum(II)(2,2’-bipyridine)(amino acid)] was reported in 1985 [17], many Pt-2,2′-bpy-based derivatives were developed and demonstrated their anticancer activities by forming DNA adducts, which lead to DNA damage coupled with induced cytotoxicity [18,19,20,21]. The 2,2′-bipyridine moiety allows the bipyridine ligand complex to form a planar configuration, and avoid conversion into a trans-isomer, while providing space to accommodate the various forms of functional bioactive molecules [22,23]. Compared to cisplatin and Pt(bpy)Cl_2_, the functional bioactive molecules added to the side chains of square-planar platinum complexes contribute to steric hindrance, resulting in stronger cytotoxic effects on cancer cells [24,25]. Therefore, the minor modifications in molecules or structure on the functional bioactive moiety of 2,2′-bipyridine complex may induce significant improvement in the desired biological activity. As the most electronegative element, the fluorine atom tends to form a covalent bond with polarity and imposes the lipophilicity characteristics in compounds with fluorocarbon-based substitutes, leading to improvements in pharmacokinetic properties, drug solubility and drug stability [26,27]. Due to its unique combination of electronegativity, size and lipophilicity, fluorine is one of the top options for substituents in drug design. Thirteen fluorine-containing pharmaceuticals have been approved by the US Food and Drug Administration (FDA) and it accounts for one-quarter of FDA-approved drugs in 2020 [28].

To increase lipophilicity, the aromatic fluorination (perfluoroalkylation) on a giving compound, such as fluorinated bipyridine platinum, exhibits a stable and amphiphilic characteristic [29] and polyfluorinated bipyridine cisplatin induces in vitro cytotoxicity against a panel of human cancer cell lines, which include MCF-7 (breast adenocarcinoma), MDA-MB-231 (breast adenocarcinoma) and A549 (lung adenocarcinoma), through the induction of S-G2/M arrest and the partial intercalation mechanism or programmed cell death within in vitro studies [30]. Moreover, the fluorine-containing bipyridine cisplatin analog dichloro[4,4’-bis(4,4,4-trifluorobutyl)-2,2’-bipyridine] platinum also displayed much stronger antiproliferative activities than cisplatin for inducing apoptosis in cancer cell lines [31]. In our current study, we assessed the in vitro cytotoxicity of a new cisplatin analogue dichloro[4,4’-bis(2,2,3,3-tetrafluoropropoxy)methyl)-2,2’-bipryridine] platinum (TFBPC) as well as its therapeutic effects on xenografts. The results revealed that TFBPC exhibited strong antiproliferative effects against several cisplatin-resistant human cancer cell lines, including MCF-7 (a human breast cancer cell line with estrogen, progesterone and glucocorticoid receptors), MDA-MB-231(a triple-negative breast cancer cell line), COLO205 (colon adenocarcinoma) and SK-OV-3 (ovarian carcinoma) and also inhibited tumor growth in animal models. When unbraiding the TFBPC-mediated antiproliferative mechanism, we found the TFBPC-DNA adducts caused increased DNA fragmentation coupled with ROS release, triggering apoptotic cell death signaling. Moreover, this therapeutic efficacy was assessed and resulted in encouraging treatment outcomes on triple-negative breast cancer in the cell line-derived xenograft (CDX) and orthotopic patient-derived xenograft (PDX) models. In this study, no significant side effects or toxicity were observed in animals. Taken together, TFBPC is a potent candidate for the treatment of patients suffering from cisplatin-resistant cancers and triple-negative breast cancers. TFBPC may provide an alternative chemotherapeutic option for patients in the future.

## 2. Materials and Methods

### 2.1. General

Cisplatin was obtained from a commercial source (Sigma-Aldrich, Burlington, VT, USA) and TFBPC was prepared by Dr. Mei-Hsiang Lin based on the literature procedure with a little modification. The chemical structures and properties of platinum-based agents used in the following study are shown in Table 1. The synthesis process is shown in Figure 1. Both chemicals for treatments were dissolved in DMSO to prepare the 80 mM stock solution and were then diluted in PBS or culture medium for animal models and in vitro studies, respectively.

Nuclear magnetic resonance (NMR) spectroscopy was recorded on Bruker DMX-500 Fourier-transform (FT)-NMR spectrometers and Agilent (600 MHz DD2 NMR) spectrometers; chemical shifts were recorded in parts per million downfield from Me4Si. High-resolution mass spectrometry (HRMS) was obtained with a Jeol JMSHX110 spectrometer. IR spectra were determined with a Perkin Elmer Frontier FTIR spectrometer. High-resolution mass spectra (HRMS) were recorded with a JEOL (JMS 700) mass spectrometer. Thin layer chromatography (TLC) was performed on Merck (Art. 5715) silica gel plates and visualized under UV light (254 nm), upon treatment with iodine vapor, or upon heating after treatment with 5% phosphomolybdic acid in ethanol. Flash chromatography was performed with Merck (Art. 9385) 40–63 lm silica gel 60. Chemicals, reagents and solvents employed were commercially available, and used as received. All reactions were carried out under an atmosphere of dry nitrogen.

### 2.2. Preparation of Compound ***6***

Compound 6 was prepared from 4,4-dimethyl-2,2-dipyridyl as reported in the literature [32,33]. In short, compound **3** (50% yield) was prepared from 4,4-dimethyl-2,2-dipyridyl via oxidation with potassium permanganate in 25% sulfuric acid and followed by reflux for 12 h. Compound **3** was further esterified in anhydrous methanol plus sulfuric acid and then refluxed for 24 h to obtain compound **4** in a 99% yield. The reduction reaction was conducted by mixing compound **4** with absolute methanol and sodium borohydride. After 30 h reflux, saturated ammonium chloride solution was added to resolve the excess sodium borohydride in the mixture. Compound **5** (75% yield) was gained after processes of evaporation and purified using column chromatography. The bromination occurred in 48% hydrobromic acid with 30 h reflux and compound **6** in yield of 30% was acquired for further TFBPC formation. **3**: ^1^H NMR (300 MHz, DMSO-d_6_) *δ* 7.96 (d, *J* = 6.0 Hz, 2H), 8.89 (s, 2H), 8.96 (d, *J* = 6.0 Hz, 2H). **5**: ^1^H NMR (300 MHz, CD_3_OD) *δ* 4.77 (s, 4H), 7.46 (d, *J* = 6.0 Hz, 2H), 8.29 (s, 2H), 8.61 (d, *J* = 6.0 Hz, 2H). **6**: ^1^H NMR (500 MHz, CDCl_3_) *δ* 4.50 (s, 4H), 7.40 (d, *J* = 4.5 Hz, 2H), 8.49 (s, 2H), 8.69 (d, *J* = 5.0 Hz, 2H).

### 2.3. Preparation of Compound ***7***

TFBPC was synthesized from compound **6** as reported in the literature [30] and modified as follows. Compound **7** (78% of yield) was prepared in a mixture of 2,2,3,3-tetrafluoro-1-propanol (0.65 mL, 7.35 mmol), anhydrous tetrahydrofuran (THF) (50 mL) and 60% sodium hydride (0.31 g, 7.64 mmol) after an hour reflux. The mixture was cooled at room temperature, followed by the addition of **6** (1 g, 2.9 mmol) and refluxing for 18 h. The reaction mixture was cooled and evaporated to remove THF. The residue was treated with water and ethyl acetate, and this solution was extracted with ethyl acetate. The organic phase of the mixture was washed with water, dried over anhydrous MgSO_4_ and the solvent was evaporated until dry. The crude products were purified using column chromatography using ethyl acetate/*n*-hexane (1:4) to produce **7** (1.0 g, 78%). **7**: ^1^H NMR (600 MHz, CDCl_3_) *δ* 3.91 (t, *J* = 12.6 Hz, 4H, H_9,9′_), 4.73 (s, 4H, H_7,7′_), 5.96 (tt, *J* = 53.1, 4.8 Hz, 2H, H_11,11′_), 7.30 (d, *J* = 4.8 Hz, 2H, H_5,5′_), 8.32 (s, 2H, H_3,3′_), 8.67 (d, *J* = 4.8 Hz, 2H, H_6,6′_); ^13^C NMR (150 MHz, CDCl_3_) *δ* 67.6 (*J*_CCF_ = 28.4 Hz, C_9,9′_), 72.7 (C_7,7′_), 109.2 (*J*_CF_ = 248.3 Hz, *J*_CCF_ = 34.6 Hz), 114.9 (*J*_CF_ = 248.3 Hz, *J*_CCF_ = 34.6 Hz), 119.3, 121.8, 146.8, 149.6, 156.1.

### 2.4. Preparation of Dichloro[4,4’-Bis(2,2,3,3-Tetrafluoropropoxy) Methyl)-2,2’-Bipryridine] Platinum (TFBPC)

Equal molar [PtCl_2_(CH_3_CN)_2_] (1.0 g, 2.87 mmol) and **7** (1.3 g, 2.87 mmol) were added to a round bottomed flask, and anhydrous DMF (20 mL) was added as the solvent. The mixture was stirred at 80 °C for 4 h and then the solvent was removed under vacuum distillation. The residue was purified with column chromatography using ethyl acetate/*n*-hexane (1:1) to produce TFBPC (0.62 g, 30%) as a yellow solid. The structure of TFBPC was analyzed using nuclear magnetic resonance spectroscopy (NMR). **TFBPC**: ^1^H NMR (500 MHz, DMSO-d_6_) *δ* 4.16 (t, *J* = 13.8 Hz, 4H, H_9,9′_), 4.85 (s, 4H, H_7,7′_), 6.65 (tt, *J* = 52.2, 5.4 Hz, 2H, H_11,11′_), 7.77 (d, *J* = 6.1 Hz, 2H, H_5,5′_), 8.43 (s, 2H, H_3,3′_), 9.39 (d, *J* = 6.1 Hz, 2H, H_6,6′_), ^13^C NMR (126 MHz, DMSO-d_6_) *δ* 67.6 (t, *J*_CCF_ = 26.9 Hz, C_9,9′_), 71.5 (C_7,7′_), 109.8 (tt, *J*_CF_ = 247.6 Hz, *J*_CCF_ = 32.8 Hz, C_11,11′_), 116.0 (tt, *J*_CF_ = 249.3 Hz, *J*_CCF_ = 25.7 Hz, C_10,10′_), 122.2, 125.6, 148.6, 152.0, 156.9; FT-IR 1625, 1558, 1429 (m, bipyridine-ring), 1230, 1204, 1100 (s, CF_2_ stretch); The NOESY data were added in the supporting file. HR-FAB (M+) C18H16F8N2O2Pt35Cl2, calcd: *m*/*z* 709.0109, found: *m*/*z* 709.0109. C18H16F8N2O2Pt35Cl37Cl, calcd: *m*/*z* 711.0079, found: *m*/*z* 711.0087. C18H16F8N2O2Pt37Cl2, calcd: *m*/*z* 713.0050, found: *m*/*z* 713.0094.

Reagents and conditions: (a) KMnO_4_/25% H_2_SO_4_, reflux 12 h, 50%, (b) H_2_SO_4_/CH_3_OH, reflux 24 h, 99%, (c) NaBH_4_/CH_3_OH, reflux 30 h, 75%, (d) 48% HBr, reflux 10 h, 30%, (e) 60% NaH, THF, 2,2,3,3-tetrafluoro-1-propanol, 78%, (f) cis-bis(acetonitrile)dichloroplatinium (II), DMF, 30%.

### 2.5. UV-Vis Spectrum of DNA-TFBPC Complex

The binding of the platinum-based agent with calf thymus DNA (ct-DNA) was assessed using UV-visible spectroscopy. In total, 1000 ng of ct-DNA (Sigma-Aldrich, Burlington, VT, USA) was incubated with 20, 40, 80, 160, 320 and 640 μM of CDDP or TFBPC in 80 μL ddH_2_O for 5 h at 37 °C. Absorption between the wavelengths of 220 nm and 500 nm was detected by measuring absorbance every 2 nm via Thermo Varioskan Flash microplate reader (Thermo Fisher Scientific Inc., Waltham, MA, USA). The maximal absorbance of DNA at 260 nm was marked to highlight the spectrum in pure DNA without chemical intercalation.

### 2.6. Gel-Based Electrophoretic Mobility Shift Assays

Plasmid DNA interacting with platinum-containing compounds was studied using agarose gel electrophoresis to separate DNA fragments dependent on sizes and structures. In total, 1 g/mL of STAT3-DN plasmid (BIOTOOLS Co., New Taipei City, Taiwan) around 4.8 kb provided by Dr. Chien-Huang Lin was incubated with an equal volume of increasing concentrations of compounds (5~40 µM) at 37 °C for 24 h. DNA–CDDP or DNA–TFBPC mixtures mixed with U-Safe™ Nucleic Acid Gel Staining Dye (Bio-genesis Technologies, Inc., Taipei, Taiwan) were examined in 1% agarose gel for 30 min at 100 volts and visualized using UVP BioDoc-It^®^ Imaging System.

### 2.7. Cell Lines

Human cancer cell lines, including MDA-MB-231, MCF-7, COLO205 and SK-OV-3, used in this study were purchased from American Type Culture Collection (ATCC, Manassas, VA, USA). All cell cultures were maintained in a 5% CO_2_-containing incubator at 37 °C. Triple-negative breast cancer cell line MDA-MB-231 and estrogen receptor-positive breast cancer cells MCF-7 were cultured in Dulbecco’s Modified Eagle Medium/Nutrient Mixture F-12 medium (DMEM/F12, Thermo Fisher Scientific Inc., Waltham, MA, USA) supplemented with 1% HyClone Penicillin-Streptomycin 100X solution (Cytiva, Marlborough, MA, USA) and 10% (*v*/*v*) fetal bovine serum (Thermo Fisher Scientific Inc., Waltham, MA, USA). COLO205 colon cancer cells and SK-OV-3 ovarian cancer cells were cultured in Roswell Park Memorial Institute (RPMI) 1640 medium (Thermo Fisher Scientific Inc., Waltham, MA, USA) containing the same supplements as described above.

### 2.8. Cell Viability Assay

Cell viability was evaluated using MTT assay. In total, 1 × 10^4^ cells were placed and cultured in a well of a 96-well microplate overnight to achieve firm attachment and then treated with platinum-containing chemotherapeutic agents either CDDP or TFBPC at the concentrations of 0.625, 1.25, 2.5, 5, 10, 20, 40 or 80 μM for another 48 h. A viable cell count was determined by adding MTT reagent (3-(4,5-Dimethyl-2-thiazolyl)-2,5-diphenyl-2H-tetrazolium bromide (MTT; SERVA, Heidelberg, Germany) at a concentration of 5 mg/mL in PBS followed by measurement of the absorbance at 570 nm for each well detected by μQuant™ universal microplate spectrophotometer (BioTek, Winowski, VT, USA). The assessment procedures were followed under the manufacturer’s instructions. The final results were represented as mean values in the same experimental group detected by a microplate reader.

### 2.9. Cell Cycle Analysis

In total, 3 × 10^5^ of MDA-MB-231 cells cultured in a 6-well plate were treated either with CDDP or TFBPC for 24 h and then were harvested using trypsin, followed by centrifugation at 1600 RPM for 5 min. The pellet was collected and fixed in 99% ice-cold ethanol and then stored in a freezer at −20 °C for further processing of staining. To label DNA, those fixed cells were rapidly washed to remove the residual fixing agent, followed by staining with DNA intercalating agent propidium iodide solution (PI) for 30 min in the dark. The PI working solution contains 0.5% Triton X-100, 15 μg propidium iodide (PI; BioVision Inc., Milpitas, CA, USA) and 10 μg RNase A (Thermo Fisher Scientific Inc., Waltham, MA, USA) in 1 mL PBS. The measurement and acquisition of DNA content were conducted using FACSVerse^TM^ Flow Cytometer (BD, Franklin Lakes, NJ, USA). Cell cycle distribution in four phases (subG1, G1, S, G2/M) was gated and analyzed in Kaluza Analysis Software (Beckman Coulter Inc., Indianapolis, IN, USA).

### 2.10. Apoptosis Detection

In total, 3 × 10^5^ MDA-MB-231 cells were first plated and cultured in a 6-well plate for 24 h and then exposed to medium, TFBPC- or CDDP-containing medium, and incubated in the dark for another 15 h, followed by apoptotic assessment using FITC-annexin V (AV) and propidium iodide (PI) double staining cell solution (Thermo Fisher Scientific Inc., Waltham, MA, USA) for 30 min. To assess the AV-Phosphatidylserine (excitation/emission maxima of 490/525 nm) and PI-DNA (excitation/emission maxima of 535/617 nm) binding, fluorescence intensity was analyzed and recorded using flow cytometer. Scatters in a two-parameter dot plot were gated into four quadrants. Proportions in the lower right (LR) and upper right (UR) quadrants were represented as the early stage and late stage in apoptotic process.

### 2.11. Reactive Oxygen Species (ROS) Detection

In total, 3 × 10^5^ MDA-MB-231 cells were seeded in a 6-well plate overnight and then incubated in medium containing 20 µM 2’,7’-Dichlorofluorescein diacetate (DCF-DA; Cayman, Ann Arbor, MI, USA) for 30 min followed by adding CDDP or TFBPC into DCF-DA-containing medium at final concentrations of 2.5, 5, 10 and 20 μM for another 8 h incubation time. DCF-DA is highly fluorescent and can label ROS for quantitative assessment using flow cytometer with excitation/emission at 485 nm/535 nm. To harvest and collect adherent and suspended cells, we first collected suspended cells from supernatant by centrifuging at 3000 rpm for 5 min at 4 °C and then adding trypsin to detach these adherent cells in the 6-well plate, followed by removing suspended cells via centrifugation. Both cell pellets after centrifugation were collected and re-suspended in PBS for subsequent quantitative analysis of ROS productions from cells treated either with CDDP or TFBPC using flow cytometer by gating the positive area in the histogram.

### 2.12. Western Blotting

To determine whether CDDP and TFBPC contribute to cell death on MDA-MB-231, cellular proteins were harvested and collected for apoptotic and autophagy signaling determination using gel electrophoresis and Western blotting. Cells after 4 or 18 h incubation with CDDP or TFBPC were lysed for protein extraction and then proteins were quantified based on the Bradford protein assay (Bio-Rad Laboratories, Hercules, CA, USA). Each sample containing 30 μg protein mixture was loaded into a well on SDS-PAGE gel and separated through electrophoresis, followed by protein determination using specific antibodies against PARP (1:1000, Cat. #9542; Cell Signaling Technology, Danvers, MA, USA), GAPDH (1:5000, Cat. GTX100118; GeneTex, Irvine, CA, USA), Bax (1:1000, Cat. ARG65612; Arigo, Glasgow, UK), Bcl-2 (1:1000, Cat. ARG55188; Arigo, Glasgow, UK) and LC3-I/II (1:1000, Cat. #12741; Cell Signaling Technology, Danvers, MA, USA). Primary antibodies were targeted by horseradish peroxidase (HRP) conjugated anti-mouse (1:5000, Cat. A9044, Sigma-Aldrich, Burlington, VT, USA) or anti-rabbit (1:5000, Cat. SI-A0545; Sigma-Aldrich, Burlington, VT, USA) secondary antibodies for visualizing immunoblots. Results were imaged by ChemiDoc system (Bio-Rad Laboratories, Hercules, CA, USA) and semi-quantification was conducted using Image Lab system (Bio-Rad Laboratories, Hercules, CA, USA).

### 2.13. DNA Fragmentation

Total DNA was extracted from cells treated with either CDDP or TFBPC for 24 h and lysed in DNA lysis buffer according to the previously published protocol [34]. DNA samples suspended in DNA lysis buffer containing 1.0 mM Tris hydrochloride (pH 7.5), 10 mM EDTA (pH 8.0), 10 mM sodium chloride and 0.5% Triton X-100. In total, 18 μL of DNA content mixed with 2 μL of 6X DNA loading dye (Geneaid, New Taipei City, Taiwan) was loaded and separated in 1.0% agarose gel under 50 volts Markers for 1 Kb DNA Ladder (MaestroGEN Inc., Hsinchu City, Taiwan) and fragmented DNA were noted and photographed by UVP BioDoc-It^®^ Imaging System.

### 2.14. Cell Line-Derived Xenograft (CDX) Model

The procedures for the establishment of breast cancer-bearing xenograft models were modified from a previous publication [35] and approved by the Taipei Medical University (TMU) Institutional Animal Care and Use Committee (IACUC No. LAC-2020-0175). Overall, 4- to 6-week-old NOD.Cg-PrkdcscidIl2rgtm1Wjl/SzJ (NSG) female mice purchased from National Laboratory Animal Center (Taipei, Taiwan) were housed in specific pathogen-free rooms with ad libitum feeding and 12 h light–dark cycle. After a week of acclimatization, 2 × 10^6^ MDA-MB-231 cells suspended in 1:1 ratio of phosphate-buffered saline plus Matrigel^®^ Matrix (Cat. 356237; Corning Inc., New York, NY, USA) were subcutaneously injected on the right flank of each mouse. Once tumor volumes reached 50 mm^3^, mice were randomly divided into four groups and were given PBS or 5 mg/kg of platinum-based agents (CDDP or TFBPC) once per week and four times in total through an intraperitoneal route. To monitor tumor growth in the models, tumor diameter was measured using an electronic caliper and values were applied on the equation: width^2^ × length × 0.52 to estimate their tumor volume. All mice were sacrificed on day 28 after treatment. Plasma, tumors and major organs (heart, liver, kidney, spleen and lung) were collected for toxicity assessment and metastasis evaluation.

### 2.15. Patient-Derived Xenograft (PDX) Model

The procedures for the establishment of breast cancer-bearing xenograft models were modified from previous publications [36,37] and approved by the Taipei Medical University (TMU) Institutional Animal Care and Use Committee (IACUC No. LAC- 2021-0546) and the Joint Institutional Review Board of Taipei Medical University (N201812005). Overall, 4- to 6-week-old NOD.Cg-PrkdcscidIl2rgtm1Wjl/SzJ (NSG) female mice purchased from BioLASCO Taiwan Co., Ltd. (Taipei, Taiwan) were housed in specific pathogen-free rooms with ad libitum feeding and 12 h light–dark cycle. Patient-derived TNBC tumors were microscopically considered for a hypervascular background and frequent mitosis features, and triple-negative results for estrogen receptor, progesterone receptor and human epidermal growth factor receptor 2 (HER2). After a week of acclimatization, patient-derived TNBC tumors dissected into 3 × 3 × 3 mm^3^ were implanted using puncture needle in fat pads between the third and fourth mammary glands for each mouse. Two weeks after tumor implantation, mice were randomly divided into four groups and were given 5 mL/kg of PBS or 2.11 mg/kg of CDDP or 5 mg/kg of CDDP or 5 mg/kg TFBPC once per week and four times in total through an intravenous route.

Tumor growth curves were generated by measuring tumor diameters and then calculating based on the same equation described above. Finally, the mice were sacrificed on day 23 after the first dose of treatments. Plasma, tumors and major organs (heart, liver, kidney, spleen and lung) were collected for toxicity assessment and metastasis evaluation.

### 2.16. Complete Blood Count (CBC) and Biochemical Analysis of Blood

Whole blood harvested from sacrificed animals was collected in EDTA-containing or heparin-containing tubes for CBC and biochemical analysis. For biochemical analysis, whole blood was centrifuged under 2000× *g* for 5 min to separate plasma from blood cells and then plasma was kept and stored in a −20 °C fridge until further analysis. Concentrations of albumin, total protein, aspartate aminotransferase (AST), alanine aminotransferase (ALT) and creatinine (CREA) in plasma, numbers of red blood cells (RBC), platelets (PLT), white blood cells (WBC) and hemoglobin (HGB) concentration in EDTA-anticoagulated whole blood were determined by medical laboratory scientists in Taipei Medical University Hospital (Taipei, Taiwan).

### 2.17. Immunohistochemistry (IHC)

Formalin-fixed and paraffin-embedded 4 μm sections of tumor and organs from cell-bearing xenograft models were stained in hematoxylin and eosin to provide an overall morphology of tissues. Tumor sections were dewaxed with UltraClear™ (J.T.Baker, Phillipsburg, NJ, USA), rehydrated in a descending series of alcohol and water and had antigens unmasked in pH 6 or pH 9 Epitope Retrieval Solution (Leica Biosystems Inc., Wetzlar, Germany) in a microwave pressure cooker. Next, sections were blocked by 3% hydrogen peroxide to inactivate endogenous peroxidase, immersed in blocking buffer to avoid background noise and stained in primary antibodies against various targets, including P53 (1:400, Cat. ab131442; Abcam, Cambridge, UK), Ki-67 (1:400, Cat. ab15580; Abcam, Cambridge, UK), cleaved caspase-3 (1:200, Cat. 9661S; Cell Signaling Technology, Danvers, MA, USA) and PD-L1 (1:100, Cat. 13684S; Cell Signaling Technology, Danvers, MA, USA). VisUCyte HRP Polymer Antibody against rabbit IgG (VC003-025; R&D Systems, Minneapolis, MN, USA) and mouse IgG (VC001-025; R&D Systems, Minneapolis, MN, USA) were targeted to primary antibodies. Liquid DAB+ system from Agilent Technologies (Santa Clara, CA, USA) was applied as a chromogen for visualizing. Slides were mounted after hematoxylin counterstaining. All images were scanned using Motic EasyScan Pro 6 pathology slide scanner (Motic China Group Co., Ltd., Hong Kong, China).

### 2.18. Preparation of Glycyrrhizic Acid Micelle (GAM)-Encapsulated TFBPC

To improve the bioavailability in in vivo studies, TFBPC was encapsulated in glycyrrhizic acid micelles according to the protocol described below. Briefly, TFBPC molecules for formulation were dissolved in acetone for a final concentration of 1 mg/mL, and 10 mg/mL of glycyrrhizic acid used as encapsulating materials was prepared in methanol. The encapsulation process was completed by mixing the same volumes of glycyrrhizic acid solution and TFBPC in a round-bottom flask. To remove all solvents in this mixture, the glycyrrhizic acid micelle (GAM)-encapsulated TFBPC was evaporated in a rotary vacuum evaporator until a gelatinous result was identified in the flask. Before being used as treatments on animal models, the gel-like mixture was reconstituted in ddH_2_O and then filtered through a 0.45 μm filter to create a homogeneous dispersion of TFBPC-loaded micelles.

### 2.19. UV-Vis Spectrum of BSA-TFBPC Complex

In total, 250 μM of bovine serum albumin (BSA) was prepared in phosphate-buffered saline (pH 7.4, PBS) and the platinum-based complex (CDDP or TFBPC) was mixed in 250 μM of BSA for final concentrations 20–640 μM. Absorption between the wavelengths of 220 nm and 500 nm was detected by measuring absorbance every 2 nm via Thermo Varioskan Flash microplate reader (Thermo Fisher Scientific Inc., Waltham, MA, USA). The maximal absorbance of BSA at 278 nm was marked to highlight the spectrum in pure BSA without chemical intercalation.

### 2.20. Statistical Analysis

Results are presented as mean ± SD and more than three independent replicates had been carried out in each experiment. One-way ANOVA, Two-way ANOVA and Tukey’s post hoc analyses were used to compare differences among groups in GraphPad Prism statistics software. The probability value (*p*-value) less than 0.05 indicated statistical significance.

## 3. Results

### 3.1. TFBPC-DNA Binding Assessment

It has been well documented that cisplatin and its deviates bind to DNA and then form DNA adducts, leading to interference with cell cycles that result in DNA damage and apoptosis of cancer cells. To evaluate the DNA chelating capability of TFBPC, ultraviolet–visible spectroscopy was applied. Each tested double-stranded DNA sample containing 25 ng/μL of calf-thymus DNA (ct-DNA) was exposed to either TFBPC or CDDP at concentrations of 40, 80, 160, 320 and 640 μM for 6 h and then subsequently had their absorbance measured from the wavelengths of 220 to 500 nm using UV-Vis spectra. The results demonstrated that the peak absorbance of pure ct-DNA was 0.6 at the wavelength of 260 nm (indicated as the solid black line in Figure 1A,C). In addition, the changes in UV-Vis absorption spectra of ct-DNA were observed in the presence of varying amounts of TFBPC. The absorption spectrum showed greater intensity with an increasing amount of TFBPC, displaying the hyperchromicity profile of ct-DNA in the presence of TFBPC (Figure 1A). The scatterplot between absorbance vs. ct-DNA in the presence of TFBPC displayed a simple linear regression with the equation y = 0.5763 + 0.00062x and an R-square value near to 1 (R^2^ = 0.9936) (Figure 1B). In contrast to TFBPC, the absorbance spectra of ct-DNA in the presence of CDDP ranged from 0 to 160 μM at 260 nm, exhibiting a hypochromic profile (Figure 1C). Moreover, it also displayed the different patterns in the scatter diagram graphs between absorbance at 260 nm vs. concentration. This resulted in a secondary polynomial progression with the equation y = 0.6011 − 0.0002484x + 0.0000003217x^2^ and R^2^ = 0.8802 (Figure 1D). Both CDDP and TFBPC induced single or double breakages of plasmid DNA, leading it to form nicked double-stranded circular or linear DNA that resulted in accelerated DNA degradation (Figure 1E).

### 3.2. Antiproliferative Assessment of TFBPC on Human Cisplatin-Resistant Cancer Cells

Results from the TFBPC-DNA binding assessment suggest that TFBPC may act as a DNA chelating agent to form DNA adducts, contributing to interference with DNA synthesis and cell proliferation. To evaluate the antiproliferation potential of TFBPC, four types of human cancer cell lines, including MCF-7 (Her2-positive breast cancers), MDA-MB-231 (triple-negative breast cancers), COLO205 (colon cancers) and SK-OV-3 (ovarian cancers), were tested. Each cell line was cultured in a medium containing either TFBPC or CDDP at various concentrations from 0.625, 1.25, 2.5, 2.5, 5, 10, 20, 40 and 80 μM for 24 h and their cell viabilities were assessed using MTT assay. The results indicated that TFPBC possessed a significantly higher cytotoxic effect on cisplatin-resistant MDA-MB-231 than CDDP. TFBPC dramatically reduced cell viability in MDA-MB-231 cells ranging from 1.25 to 80 μM (Figure 2A). TFBPC and CDDP at 20 μM displayed the maximal differential (~78%) in cell viability with 88% and 10% reductions, respectively. For cisplatin-resistant MCF-7 cells, both CDDP and TFBPC displayed a little inhibitory effect on cell viability. At 80 μM, CDDP and TFBPC delivered an antiproliferative effect with a 19% and 28% reduction, respectively (Figure 2B). TFBPC exhibited a higher antiproliferative effect on COLO205 at concentrations ranging from 10 to 40 μM, compared to CDDP (Figure 2C). CDDP and TFBPC at 20 μM imposed proliferative inhibition on COLO205 with 23% and 72% reductions, respectively. Compared to the other three cell lines, SK-OV-3 cells were most sensitive to TFBPC at low doses of 0.625, 1.23 and 2.5 μM with 29, 54 and 78% reductions, respectively (Figure 2D). The IC_50_ values of CDDP and TFBPC against MCF-7, MDA-MB-231, COLO205 and SK-OV-3 cell lines were listed as shown in Table 2. Both MDA-MB-231 and MCF-7 breast cancer cells displayed drug tolerance to CDDP with IC_50_ greater than 80 μM. However, MDA-MB-231 was relatively sensitive to TFBPC with IC_50_ at 3.86 μM.

### 3.3. Cell Cycle Arrest Analysis of MDA-MB-231 Exposed to TFBPC

Based on the previous study, TFBPC can interact with double-stranded DNA and inhibit proliferation of cisplatin-resistant MDA-MB-231 cells. This evidence suggests that TFBPC might interfere with cell cycle progression of breast cancer cells, leading to failure in cell division. To examine our hypothesis in TFBPC-mediated cell cycle arrest, MDA-MB-231 cells were first starved for 24 h to synchronize cell division and then incubated in medium containing various concentrations of CDDP or TFBPC at 2.5, 5 to 10 μM for another 24 h prior to DNA content/cell cycle analysis. After completion of 24 h drug exposure, MDA-MB-231 cells were fixed in 99% ethanol to stop the cell-division cycle and then propidium iodide (PI) staining was performed, followed by flow cytometer analysis. The DNA content (%) in each stage of cell cycles, including the sub G1 phase (colored in orange), G1 phase (green), S phase (purple) and G2/M phase (pink), was quantified and listed in DNA content histograms (Figure 3A). The results revealed a statistically significant increase in DNA content in the S phase of cells exposed to CDDP (Figure 3B). Compared to the DNA content in cells without exposure to CDDP as 14.75%, an increased accumulation of DNA content in cells treated with CDDP at 2.5, 5 and 10 μM is 26.8, 32.27 and 39.65%, respectively. However, it displayed a different pattern in cells exposed to TFBPC with significantly increased DNA accumulation in the sub G1 phase (Figure 3C). Compared to cells without exposure to TFBPC (8.71%), the DNA content of cells exposed to TFBPC at 2.5, 5 and 10 μM in sub G1 is 14.8, 23.14 and 27.15%, respectively.

### 3.4. TFBPC-Mediated Apoptotic Cell Death Program

Based on the evidence of combined TFBPC-mediated antiproliferation and cell cycle arrest, we speculated TFBPC may initiate a cell death program. To detect the apoptotic cells, FITC-conjugated Annexin V staining plus flow cytometry was applied. This revealed that CDDP failed to induce MDA-MB-231 cells to undergo apoptosis, as depicted in Figure 4A,C. However, it clearly showed that TFBPC caused cells to undergo an apoptotic fate, as depicted in Figure 4A, and TFBPC-induced apoptosis of MDA-MB-231 was in a concentration-dependent manner as seen in Figure 4B. Furthermore, DNA fragmentation confirms TFBPC-induced apoptosis of MDA-MB-231 (Figure 4C). Considering the central roles of reactive oxygen species (ROS) in apoptosis induction, the assessment of ROS detection was performed, and the results revealed an apparent increase in ROS production from cells exposed to TFBPC (Figure 4D). To determine the molecular mechanisms associated with TFBPC-induced cell death, apoptotic and autophagy markers were characterized and investigated using Western blotting (Figure 4E). The results revealed LC3 I conversion to LC3 II (a typical autophagy marker) (Figure 4G) and the formation of the cleaved form of PARP (Poly [ADP-ribose] polymerase 1) (Figure 4F) plus an increase in Bax/Bcl-2 ratio (apoptotic markers), suggesting that TFBPC induced ROS production coupled with autophagy and apoptotic cell stress.

### 3.5. In Vivo Therapeutic Assessments of TFBPC

It has been demonstrated that TFBPC causes cisplatin-resistant TNBC cell cycle arrest coupled with death in current in vitro studies. To extend our knowledge and clinical application in the future, therapeutic assessments were performed in two animal models: a cell line-derived xenograft (CDX) and a patient-derived xenograft (PDX). The cisplatin-resistant MDA-MB-231 cells were subcutaneously implanted into the right flank of NSG mice (NOD Scid Gamma) to establish xenograft models for therapeutic assessment of TFBPC. Mice were intraperitoneally administrated with PBS (5 mL/kg), CDDP (2.11 or 5 mg/kg) or TFBPC (5 mg/kg) weekly after 3-week cancer cell implantation. Due to the distinct difference in molecular weight between CDDP and TFBPC, two doses of CDDP containing 2.11 or 5 mg/kg were injected into mice to equally match the number of molecules or to dose by weight of TFBPC at 5 mg/kg. Each animal received four instances of drug injection at days 0, 7, 14 and 21 (marked as black arrows in Figure 5A) in the 28-day schematic schedule for therapeutic evaluation. Animals were sacrificed on day 28 and tumors were resected, while major organs and blood were collected. The results indicated tumors derived from the animal without any therapeutic treatment (PBS only) had tumor volumes of 1412.72 ± 120.85 mm^3^, which were the biggest average tumor volume (*p* < 0.0001) among tumors from xenografts that underwent administration of CDDP or TFBPC. The average tumor volume in each group of mice treated with CDDP at 2.11, 5 mg/kg or TFBPC at 5 m/kg were around 821.66 ± 133.75, 540.23 ± 133.20 and 413.03 ± 87.50, respectively (Figure 5A). The representable tumor images from each group of animals were measured and compared as depicted in Figure 5B. Moreover, the tumors resected from xenografts treated with TFBPC displayed ~0.3 g in weight, which is the lightest among all groups (Figure 5C). This was consistent with the results from the tumor volume comparing studies. No substantial changes in body weight of each animal and its major organs prior to in vivo therapeutic studies were observed. Blood was drawn from the sacrificed mice and was followed by performing hematological assessments, including hematological testing and biochemistry profiling. All the acquired parameters were listed in Table 3. TFBPC as well as CDDP did not alter changes in biochemical parameters, which include the amount of total albumin, total protein, creatinine (CREA), alanine aminotransferase (AST) and aspartate aminotransferase (ALT), and all the values stayed in the normal physiological range. In addition, no significant changes in the amounts of white blood cells and red blood cells were observed and the measurement of these parameters was good and remained within the range of the normal physiological condition.

To confirm the therapeutic effects of TFBPC in CDX, another assessment was performed on the orthotopic PDX model (Figure 5G) under the same experimental schematic protocol of the CDX model described in the previous paragraph. The result indicated the average tumor volume of PDX displayed the smallest size in volume (Figure 5E) and weight (Figure 5F) after injection of 5 mg/kg TFBPC. While there was no statistical difference in the average tumor volume between the control and CDDP-treated animals, there was a somewhat declining trend with CDDP treatments and their dosage (Figure 5D). The resected tumor images or orthotopic tumors of PDX were depicted in Figure 5E,F. 

In addition, there was an interestingly strong correlation between chemotherapeutic and prognostic markers in the IHC staining of tumor tissues. To elucidate expressions of putative prognostic biomarkers, tumors from cell-derived xenografts were preserved for paraffin-embedded tissue sections followed by IHC staining. Several markers of interest were examined and detected, including P53, Ki-67, cleaved caspase 3 and PD-L1, which are representative of cell proliferation, apoptosis-related cell death and immune checkpoint for preventing excessive immune responses, respectively (Figure 5H). The biopsied tumor samples from the control animals without any therapeutic treatments showed the highest expressions of P53, Ki-67 and PD-L1 coupled with the lowest expression of cleaved caspase 3 among all groups. Compared to the lower dose of CDDP (2.11 mg/kg), tumors resected from mice treated with CDDP at 5 mg/kg expressed lower intensity in P53, Ki-67 and PD-L1 and higher intensity of cleaved caspase 3. Moreover, tumor biopsy samples from animals treated with TFPBC showed the lowest level in expressions of P53, Ki-67 and PD-L1 couple with the highest expression of cleaved caspase 3. These data suggest that TFPBC exhibited the best therapeutic efficacy with the highest expression of apoptosis marker cleaved caspase 3 and the lowest expression of tumor progressive markers P53 and Ki-67, and immune checkpoint protein PD-L1.

In addition, TFBPC-induced toxicity and side effects were also explored in this study to evaluate possible detrimental effects. To compare a comprehensive picture of the microanatomy of organs and tissues between samples without or with various treatments, the major organs, such as heart, kidneys, liver, lungs and spleen, were acquired from sacrificed animals for the subsequent process of HE staining. No obvious pathophysiological changes or increased numbers of infiltrated leukocytes were observed in all biopsy samples (Figure 5I).

There were no significant changes in body weight among each group of PDXs. TFBPC and CDDP did not cause weight changes in most major organs except for the spleen (Table 4). Compared to the control and TFBPC-treated mice, a statistically significant reduction in spleen weight in animals treated with CDDP was noticed. In the results of hematological testing, there were no alterations in the amounts of hemoglobin (HGB) and platelet count in animals treated with CDDP or TFBPC. However, CDDP at 5 mg/kg reduced the RBC and WBC counts. This marginal reduction was below the normal parameter range (** p* < 0.05). The total white counts in animals that received TFBPC are significantly lower than one in the control PDX animal, but it still stayed within the normal range of WBC count.

## 4. Discussion

CDDP was the first platinum-based drug approved by the FDA for cancer therapy in 1978 and has been used in chemotherapeutics for various forms of cancer for decades. Due to its adverse side effects and drug resistance development, scientists continue to work on improving its therapeutic effects and safety through the structural re-modification of cisplatin. Based on the molecular structure, functional groups on the squared planar complex allow cisplatin to chelate in double helix structures of nucleic acids. Therefore, alterations or modified substations on these functional groups may improve drug tumor-targeting efficiency coupled with decreased minimal dosage required for therapeutic efficacy, leading to lower cytotoxic effects exerted on patients [22]. It has been reported that a higher sterically hindered effect is believed to improve DNA binding affinity and decrease cytotoxic activity [21,39]. The platinum complex with aromatic rings not only extends the steric interference but also provides more space for substitution reactions [40,41], including substituted pyridine or bipyridine which are attributed with greater DNA targeting abilities [19,42,43,44,45] and anticancer effects than cisplatin [46,47,48,49]. In addition, modified bipyridine ligands with bioactive groups on platinum-based compounds ameliorate its stability in DMSO and cytotoxicity in cell culture [23]. Fluorine-containing moieties are highlighted in drug discovery, and currently almost 25% of FDA-approved drugs contain at least one fluorine atom [27,28]. The previous publication shows that the additional trifluorinated ligand increases the apoptosis-dependent cell death and G1 arrest in breast cancer and lung cancer [31]. Improved antitumor efficiency is considered in fluoro-pharmaceuticals because of the increased hydrophobicity and fluorophilicity [29], and the numbers of fluorine elements in polyfluorinated analogues are critical to their cytotoxicity as described by Chang [30]. Platinum analogues with four fluorine atoms induced UV-visible absorption shifts towards up and right, and inhibited tumor growth through cell cycle arrest at the S phase. Slight structural differences change the main principle behind cancer models.

In the supplementary results, we had evaluated the in vitro cytotoxicity of two additional derivatives bearing bipyridine-PtCl_2_ structure, including (4,4-3F) PtCl_2_ (Appendix A) and (4,4-5F) PtCl_2_ (Appendix A). The structures in each compound were depicted as shown in Appendix A. We assessed the viability of each compound on three cisplatin-resistant cancer cell lines, including MDA-MB-231, COLO205 (colon cancer) and HTC-15 (colon cancer) cells. The results indicated all bipyridine-PtCl_2_ containing compounds significantly inhibited cell viability of MDA-MB-231 cells (Appendix A). TFBPC and (4,4-3F) PtCl_2_ were also effective in inhibiting the cell viability of COLO205 cells (Appendix A). TFBPC and (4,4-5F) PtCl_2_ also statistically reduced cell viability of HCT-15 cells (Appendix A). In general, the Appendix A mentioned above indicated that the bipyridine-PtCl_2_-containing compounds, such as TFBPC derivatives, exhibited significant cytotoxic effects on various forms of cancer cell lines by inhibiting cell viability. Compared to (4,4-3F) PtCl_2_ and (4,4-5F) PtCl_2_, TFBPC displayed a greater superiority and effectiveness in inhibiting cell viabilities on all three types of cancer cell lines. Based on these in vitro results, TFPBC was selected for further investigation to evaluate the therapeutic outcomes of breast cancers, and it was fully examined in this study to determine its mechanism using in vitro and in vivo models. The polyfluorinated compounds displayed a complicated complex in structural chemistry. They have been investigated for their intermolecular interactions and crystal packing to interpret their crystallographic characteristics. The low solubility of those fluorine-containing drugs will be studied in drug delivery design [50,51,52].

UV absorption spectra of DNA bases ranges from 200–350 nm and the maximal absorbance of pure double-stranded DNA in spectrum absorption occurs at 260 nm. Therefore, the UV absorption spectrum is a useful tool to assess and determine the drug–DNA interactions based on the absorbance shift [53]. Various patterns in the change of UV absorption are correlated to the binding properties between drugs and double-stranded DNA. The three common shifts in UV absorbance induced by DNA chelating agents, which include hyperchromicity, hypochromicity and bathochromic shift (red shift), are observed. The hyperchromic effect is referred to as a marked increase in absorbance of DNA upon denaturation induced by DNA-binding drugs [54]. In contrast, hypochromicity is described as a drug-induced decrease in the ability to absorb light. In addition, bathochromic shift refers to a change in spectral band position in UV/visible light absorption of DNA to a longer wavelength (red shift or lower frequency). Compared to the results from UV absorbance, CDDP and TFBPC displayed different UV/visible light absorbance spectrums. There is a positive correlation between the UV/visible light absorption of DNA and the concentration of TFBPC with a regression correlation coefficient R^2^ = 0.9936, indicating a TFBPC-induced hypochromic effect on single DNA from a double strand of DNA. In contrast, CDDP induced a hypochromic effect on DNA absorbance at a CDDP concentration less than 160 µM. At higher concentrations of CDDP (320 and 640 µM), the bathochromic shift was observed. These results indicated TFBPC and CDDP applied different DNA-binding patterns resulting in the distinct UV/visible light absorbance spectrums. The previous studies indicated CDDP bound to DNA and then formed either interstrand or intrastrand cross-links with hypochromic effects as well as bathochromicity [55,56]. These data are consistent with our results when the cisplatin concentration is less than 160 µM. It has been well known that the red shift and hypochromism in the absorption spectrum results from CDDP-purines binding (guanosine and adenosine), which alters the interactions between hydrogen bonds and then disrupts the Watson−Crick base pairs in ct-DNA [57]. Both TFBPC-DNA and CDDP-DNA adducts caused circular plasmid DNA damage to form nicked or linear DNA products.

To evaluate the cytotoxicity of TFPBC on cancer cells, several cisplatin-resistant cancer cell lines were investigated. The results demonstrated that MCF-7 was also resistant to TFBPC. However, the other cisplatin-resistant cancer cell lines including MDA-MB-231, COLO205 and SK-OV-3 were highly sensitive to TFPBC with IC_50_ values at 3.86, 11.48 and 1.92, respectively (Table 2). This indicates that TFBPC may emerge as a new alternative chemotherapeutic agent for cisplatin-resistant cancers. Among these three cell lines, MDA-MB-231 was the most sensitive to TFPBC. Therefore, we characterized TFPBC-mediated molecular mechanisms on inhibiting cell proliferation of the triple-negative breast cancer cell line MDA-MB-231. We found the DNA contents in MDA-MB-231 cells exposed to TFBPC at 5 and 10 μM were predominantly abundant in the sub G1 phase, suggesting that TFBPC induced large amounts of DNA fragmentations. This result was consistent with the TFBPC-mediated antiproliferation assessments of MDA-MB-231 in Figure 2A. Instead of sub G1 accumulation, the DNA content of cells treated with CDDP accumulated in the S phase and was coupled with a reduction in the G1 phase, suggesting that cells were arrested in the S phase, which is related to the activation of apoptosis and/or survival. This implies TFBPC targeted and damaged DNA quickly without preference on a particular cell cycle or stage. This shows that TFBPC-DNA adducts contributed to the occurrence of apoptotic cell death events, such as increased phosphatidylserine (PS) expression detected by annexin V (an early event in apoptosis) and DNA laddering pattern (a key event of apoptosis) shown in Figure 4B,C. In contrast with TFBPC, there is no sign of cisplatin-induced PS expression or DNA fragmentation in MDA-MB-231. We also found TFBPC mediated a significant release of ROS from cells in a dose-dependent manner. To understand the TFBPC-mediated cell death molecular mechanism, several intermediate molecules associated either with apoptosis or autophagy were harvested and investigated using electrophoresis combined with Western blotting. We found decreased expressions of the full-length form of PARP coupled with increased production of cleaved PARP in MDA-MB-231 cell post-treatment of TFBPC, suggesting the TFBPC-induced apoptotic signals in MDA-MB-231. Moreover, the conversion of LC3 I to LC3 II was observed in cells treated with TFBPC, indicating TFBPC induced autophagy in MDA-MB-231. The ratio between autophagosomal membrane-associated form of LC3 II and cytosolic truncated form of LC3 I is used for determining the occurrence of autophagy [58]. Autophagy is a natural process of self-degradation that saves or balances the energy consumption of eukaryotic cells for survival under extreme conditions. The ultimate role of autophagy in cell fate is ambiguous. It may come with pro-death and pro-survival characteristics, which determine therapeutic outcomes and drug resistance [59,60]. The previous studies indicated that prolonged exposure to cisplatin triggered drug resistance through the inhibition of autophagy [61,62]. In breast cancer models, cisplatin suppressed cell growth and cell cycle progression in an autophagy-dependent manner [63].

TFBPC significantly reduced tumor growth in breast cancer cell-derived xenografts and patient-derived xenografts, suggesting that TFBPC is a potentially effective chemotherapeutic candidate. To analyze therapeutic outcomes using histological assessments, several key markers strongly associated with tumor progression and therapeutic outcome, including P53, Ki-67, cleaved caspase 3 and PD-L1, were investigated using immunohistochemistry. It has been demonstrated that Ki-67, when used as a proliferation index is highly expressed in patients with worse disease-free survival (DFS) and overall survival (OS) [64,65] and the same is true for an increased expression of P53 [66]. In addition, PD-L1 is an immune checkpoint protein for assisting cells in avoiding immunosurveillance. Therefore, increased expressions of PD-L1 may serve as a negative prognostic marker [67,68]. Our results revealed TFPBC treatment delivered great therapeutic efficacy and therapeutic outcome with significantly decreased expression of P53, Ki-67 and PD-L1 coupled with increased expression of cleaved caspase 3 in the tissues exposed to TFPBC. These histological results indicated TFBPC possesses great potential as a therapeutic candidate for treatments of triple-negative breast cancers.

The therapeutic efficacy of TFBPC has been evaluated in CDX and PDX animal models in this study. Besides the difference in the tumor origin between two animal models, there is another difference in how the tested drug was delivered to animals. CDDP and TFBPC were intraperitoneally administered into mice in the CDX, but intravenously injected into animals in the orthotopic PDX model. Although the administration of chemotherapeutics differed in CDX and PDX, TFBPC-mediated therapeutic efficacy on TNBCs remained the most effective with the best therapeutic outcome among both groups, regardless of whether they were CDX or PDX animal models. This indicates that the lipophilic TFPBC was absorbed efficiently from the peritoneal cavity and retained functional stability in the bloodstream for achieving its therapeutic purpose. This suggests that the lipophilicity of TFPBC increased its passage through tumor cell membranes and so required a reduction in drug dosage, ultimately resulting in minimizing the side effects. Moreover, the encapsulation of TFBPC in glycyrrhizic acid micelles (GAM) for improving bioavailability was also formulated and tested in the CDX animal model. The results revealed that the average tumor volume at day 49 in xenografts exposed to GAM-encapsulated TFBPC was ~500 mm^3^ and ~400 mm^3^ in animals treated with TFBPC (seen in Appendix A). Compared with the tumor size of mice that received no treatments, both TFBPC and GAM-encapsulated TFBPC drugs statistically significantly reduced tumor volume. Encapsulated TFBPC in GAM provided better therapeutic effects compared to TFBPC, but not at a statistically significant level, suggesting that TFBPC possesses a quality of bioavailability even without packaging it into micelles.

Because of the hydrophobic characteristics in TFBPC, the transportation route of this hydrophobic complex is inadequate when administered via intravenous infusion; however, TFBPC performed with an efficacious response in our PDX model via tail vein injection. The consequence of tumor suppression in animal models is uncertainly interpreted from the structure of TFBPC. In the development of targeted delivery of antitumor therapies, abundant plasma proteins, such as albumin and α -1-acid glycoprotein, play as impressive carriers for endogenous and exogenous substance transport in the bloodstream. The interaction of plasma albumin and chemotherapeutic drugs achieves lower drug clearance and side effects by decreasing the unbound fraction of drugs in the systematic circulation [69]. BSA shares similar structure with human serum albumin with a 75.6% sequence homology, possesses dynamical properties and exhibits low procurement cost and ready availability [70]. Therefore, the protein-binding prolife of TFBPC with bovine serum albumin (BSA) was evaluated supplied using UV-visible spectroscopy in our Appendix A. The results indicated the intensity of UV absorbance at 278 nm was positively correlated to the concentration of TFBPC with R^2^ = 0.9554 in the presence of BSA-containing solution (Appendix A). Compared to TFBPC, the cisplatin-BSA binding correlation showed a different profile as indicated in Appendix A. Moreover, no red or blue shifts were observed in the absorbance–wavelength plot. The maximal absorbance of BSA at 278 nm results from the aromatic rings in tyrosine and tryptophan, and the shift in the peak absorbance reflects changes in the polarity and the microenvironment around tyrosine and tryptophan residues [71]. These results suggest that the conformational change of BSA resulted from the formation of a ground-state BSA-TFBPC complex without disturbing the electronic transition in amino acids of BSA [72]. Based on this preliminary UV-visible spectroscopic data, BSA likely interacted with TFBPC, leading to shifts in the absorbance of BSA. To determine TFBPC-induced changes in the secondary structure of BSA, assessments of other spectroscopic properties such as circular dichroism (CD), synchronous fluorescence, fluorescence quenching spectra and pharmacokinetic profiles of TFBPC should be used for a more detailed evaluation.

## 5. Conclusions

The ultimate purpose of this study is to design and formulate a new cisplatin derivative TFBPC with good therapeutic effects coupled with enhanced lipophilicity for increased drug absorption and passage over the tumor membrane for reducing side effects. The therapeutic effect of TFBPC was assessed in the two CDX and PDX animal models, and the results revealed a great therapeutic efficacy compared to the control and CDDP-treated animals in both animal models. There was no significant difference in therapeutic efficacy between intraperitoneal and intravenous injections of TFBPC into animals. Moreover, the data from the histological analysis showed that TFPBC treatment led to an improved therapeutic outcome and positive prognostic marker changes by reducing the expressions of P53, Ki-67 and PD-L1 combined with increased expression of cleaved caspase 3. Taken together, these results demonstrated that TFPBC is a good therapeutic candidate for treatments of patients suffering from TNBCs or cisplatin-resistant cancers.

We also investigated TFBPC-mediated molecular mechanisms associated with cytotoxic effects on cisplatin-resistant MDA-MB-231 cells. We found that TFBPC bound to DNA and then caused DNA damage, triggering DNA fragmentation, increasing ROS production and triggering the cell death programs for apoptosis and autophagy. However, CDDP failed to induce the formation of the cleaved form of PARP and conversion of LC3 II from LC3 I, indicating that CDDP could not induce apoptotic and autophagy signaling in MDA-MB-231.

In summary, we were successful in synthesizing the potential cisplatin-containing derivative TFBPC and assessing its therapeutic effect coupled with cytotoxic molecular mechanisms in eradicating triple-negative breast cancers in CDX and orthotopic PDX animal models. Moreover, the TFBPC treatment resulted in positive therapeutic and prognostic markers on the specimen as well as good parameters in hematological and biochemical measurements. In conclusion, TFBPC can emerge as a potential therapeutic candidate for treatments of cisplatin-resistant cancers and triple-negative breast cancers.

## Data Availability

Not applicable.

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
