# Peer review of "Preclinical Therapeutic Assessment of a New Chemotherapeutics [Dichloro(4,4’-Bis(2,2,3,3-Tetrafluoropropoxy) Methyl)-2,2’-Bipryridine) Platinum] in an Orthotopic Patient-Derived Xenograft Model of Triple-Negative Breast Cancers"

_pharmaceutics, 2022, doi:10.3390/pharmaceutics14040839_

Round 1
Reviewer 1 Report
The article entitled: Preclinical therapeutic assessment of a new chemotherapeutics 2[dichloro(4,4'-bis(2,2,3,3-tetrafluoropropoxy) methyl)-2,2'-bi- 3pryridine) platinum] in an orthotopic patient-derived xenograft 4model of triple negative breast cancers.
Which concern:
a polyfluorinated bipyridine-modified cisplatin analogue, dichloro[4,4'-bis(2,2,3,3-tetrafluoro- propoxy)methyl)-2,2'-bipryridine] platinum (TFBPC), was synthesized. TFBPC displayed the superior effects to inhibit proliferation of several cisplatin-resistant human cancer cell lines including MDA-MB-231 breast cancers, COLO205 colon cancers and SK- 32OV-3 ovarian cancers. TFBPC bound to DNA and formed DNA crosslink resulted in DNA degradation, triggering the cell death program through the PARP/Bax/Bcl-2 apoptosis and LC3-related autophagy pathway. Moreover, TFBPC significantly inhibited tumor growth in both animal models. Furthermore, the biopsy specimen from TFBPC-treated xenografts revealed decreased expressions of P53, Ki-67 and PD-L1 coupled with higher expression of cleaved caspase 3, suggesting TFBPC-treatment were effective and came with good prognostic indications. No significant pathological changes were observed in hematological. TFBPC is a potential candidate for treatments of patients suffered with TNBCs as well as other cisplatin-resistant cancers.
Comments
In the Preparation of Dichloro[4,4'-Bis(2,2,3,3-Tetrafluoropropoxy) Methyl)-2,2'-Bipryridine] Plat- inum (TFBPC)
Missed the data of the compound and all this data should be in the supporting file.
The method of synthesizing compound 7 is not written in the full experiment data(how many mols were used, the solvent, …).
The spectral data of the compound is not defined (1HNMR,13CNMR, MASS, IR) and it should be in the supporting file, and new tools like (NOE NMR) is missed.
As the compound will use as a drug, it should be known if it is, cis or trans as this has a big difference in the activity.
How is the purity of the compound done? Is by column chromatography or other tools and which solvents are used.
All the other sections are written in a good way.
The tables and figures are in a good manner with their explanation.
The paper was written from the journal perspective.
The references need more attention and missing the name of the Database for each reference.
The paper needs a minor check on the spelling and grammar.
e.g. In the introduction line, 58 (Many kinds of chemotherapeutic) need to change to (Many kinds of chemotherapy).
Line 93 (has been developed and demonstrated for their anticancer activities by formation) should be (have been developed and demonstrated for their anticancer activities by the formation).
Line 99 (complexes contributes to) should be (complexes contribute to)….etc
Author Response
Thanks for your comments and suggestions. We revised this manuscript to improve the quality of this study. The revised manuscript has been edited by an English-speaking native. In addition to the response in the attached file, we updated more information in the materials and methods (Section 2.2 to 2.4) and the supplementary figures (Fig. S1 to S10). Please see the detail in the revised manuscript and the supplementary files.

Reviewer 2 Report
In this manuscript by Kan et al., the authors presented the synthesis of a new cisplatin derivative, named TFBPC, and carried out different biological assays to prove that the molecule had good therapeutic effects in the treatment of breast cancer. Despite the reputable profile of cisplatin as a medication used in oncology that backs up the synthesis of this derivative, according to the reviewer, the scope of this study is too narrow, as only one derivative was synthesized and studied throughout the manuscript. What about other derivatives bearing the same bipyridine scaffold? Have the authors investigated those with other substituents, or other positions of the substituents in the aromatic rings? Could they be synthesized by the same scheme the authors presented here? Would studying such molecules have an impact on the results and/or discussion of this manuscript, as well as on future research? Synthesizing, for example, a series of cisplatin derivatives bearing the same bipyridine scaffold, and investigating their therapeutic profiles (to see if there is any difference and put forward useful suggestions) may be interesting from both the medicinal chemistry and the pharmacology points of view.
Besides, the reviewer remarks that the manuscript is inundated with grammatical errors, spelling mistakes, and formatting issues. Most of them could have been easily avoided if the authors had double-checked their manuscript more carefully before submission. Just to cite a few:
- A lot of subject-verb agreement errors, e.g. in line 54 (should be "have been developed", not "has been developed"), or in line 385 (should be "bind to", not "binds to").
- Serious spelling errors that change the meaning of the sentence, or render the sentence meaningless, e.g. in line 71 (should be "serving as", not "severing as"), or in line 400 (should be "ranged from", not "raged from").
- Errors related to word forms, e.g. in line 48 (should be "the most intriguing", not "the most intrigue").
- The use of the article "the" is inappropriate in a lot of instances in this manuscript. Sometimes it is used where it should not be, and vice versa. For example, in lines 28, 31, 36, 42, 49, 76 and many others.
- Other errors, e.g. in line 51 (should be "who died of cancer", the word "who" is missing), in line 42 (should be "Taken together", not "Taking together"), or in line 56 (should be "cannot" instead of "can't", contractions like this should be avoided in academic writing).
There are a whole lot of other errors. Their presence impedes reading and may make readers annoyed when they read the manuscript. The authors should ask a native speaker of the English language or an academic writing coach to double-check their manuscript the next time they submit one.
For the reasons mentioned above, it is unfortunate that the reviewer has to kindly suggest the rejection of this manuscript, and hopes that its quality will be improved if the authors decide to resubmit it somewhere else.
Author Response
Thanks for your comments and suggestions. We revised this manuscript to improve the quality of this study. The revised manuscript has been edited by an English-speaking native. In addition to the response in the attached file, we updated more information in the supplementary figures (Fig. S11). Please see the detail in the revised manuscript and the supplementary files.

Reviewer 3 Report
The authors present an interesting study, demonstrating the antitumor effects of cisplatin derivative TFBPC in vitro and in vivo. The manuscript is well-structured; the introduction and discussion provide sufficient information regarding the current advances in platinum-based antitumor drugs.
Did the authors evaluate protein-binding profile of TFBPC (such as HSA and AGP)?
A comparison in pharmacokinetic profiles between intraperitoneally and intravenously administered non-encapsulated compound would also be very useful.
Adding this additional data would significantly improve the study.
The procedure for handling and maintenance of patient-derived tumors should be described in more detail. The authors refer to a previous publication, but that publication also does not adequately describe the experimental setup.
Author Response
Thanks for your comments and suggestions. We revised this manuscript to improve the quality of this study. The revised manuscript has been edited by an English-speaking native. In addition to the response in the attached file, we updated more information in the establishment of PDX animal model and BSA binding profile in the materials and methods (Section 2.15 and 2.19, respectively) and the results was attached in the supplementary figures (Fig. S13). Please see the detail in the revised manuscript and the supplementary files.

Round 2
Reviewer 2 Report
I have read this revised version as well as the authors' responses, and am pleased with the improvements made by the authors. The manuscript is now much better and I therefore recommend it for publication.
Reviewer 3 Report
The authors have significantly improved the paper.